# Construction of Enzyme-Responsive Micelles Based on Theranostic Zwitterionic Conjugated Bottlebrush Copolymers with Brush-on-Brush Architecture for Cell Imaging and Anticancer Drug Delivery

**DOI:** 10.3390/molecules27093016

**Published:** 2022-05-07

**Authors:** Fangjun Liu, Dun Wang, Jiaqi Wang, Liwei Ma, Cuiyun Yu, Hua Wei

**Affiliations:** 1State Key Laboratory of Applied Organic Chemistry, Key Laboratory of Nonferrous Metal Chemistry and Resources Utilization of Gansu Province, College of Chemistry and Chemical Engineering, Lanzhou University, Lanzhou 730000, China; liufj18@lzu.edu.cn (F.L.); malw@lzu.edu.cn (L.M.); 2Hunan Province Cooperative Innovation Center for Molecular Target New Drug Study & Department of Pharmacy and Pharmacology, University of South China, Hengyang 421001, China; 20202012110818@stu.usc.edu.cn (D.W.); jiaqiwang@stu.usc.edu.cn (J.W.)

**Keywords:** brush-on-brush, zwitterionic, enzyme-responsive, theranostic, FRET

## Abstract

Bottlebrush copolymers with different chemical structures and compositions as well as diverse architectures represent an important kind of material for various applications, such as biomedical devices. To our knowledge, zwitterionic conjugated bottlebrush copolymers integrating fluorescence imaging and tumor microenvironment-specific responsiveness for efficient intracellular drug release have been rarely reported, likely because of the lack of an efficient synthetic approach. For this purpose, in this study, we reported the successful preparation of well-defined theranostic zwitterionic bottlebrush copolymers with unique brush-on-brush architecture. Specifically, the bottlebrush copolymers were composed of a fluorescent backbone of polyfluorene derivate (PFONPN) possessing the fluorescence resonance energy transfer with doxorubicin (DOX), primary brushes of poly(2-hydroxyethyl methacrylate) (PHEMA), and secondary graft brushes of an enzyme-degradable polytyrosine (PTyr) block as well as a zwitterionic poly(oligo (ethylene glycol) monomethyl ether methacrylate-co-sulfobetaine methacrylate) (P(OEGMA-*co*-SBMA)) chain with super hydrophilicity and highly antifouling ability via elegant integration of Suzuki coupling, NCA ROP and ATRP techniques. Notably, the resulting bottlebrush copolymer, PFONPN_9_-*g*-(PHEMA_15_-*g*-(PTyr_16_-*b*-P(OEGMA_6_-*co*-SBMA_6_)_2_)) (P_2_) with a lower MW ratio of the hydrophobic side chains of PTyr and hydrophilic side chains of P(OEGMA-*co*-SBMA) could self-assemble into stabilized unimolecular micelles in an aqueous phase. The resulting unimolecular micelles showed a fluorescence quantum yield of 3.9% that is mainly affected by the pendant phenol groups of PTyr side chains and a drug-loading content (DLC) of approximately 15.4% and entrapment efficiency (EE) of 90.6% for DOX, higher than the other micelle analogs, because of the efficient supramolecular interactions of π–π stacking between the PTyr blocks and drug molecules, as well as the moderate hydrophilic chain length. The fluorescence of the PFONPN backbone enables fluorescence resonance energy transfer (FRET) with DOX and visualization of intracellular trafficking of the theranostic micelles. Most importantly, the drug-loaded micelles showed accelerated drug release in the presence of proteinase K because of the enzyme-triggered degradation of PTyr blocks and subsequent deshielding of P(OEGMA-*co*-SBMA) corona for micelle destruction. Taken together, we developed an efficient approach for the synthesis of enzyme-responsive theranostic zwitterionic conjugated bottlebrush copolymers with a brush-on-brush architecture, and the resulting theranostic micelles with high DLC and tumor microenvironment-specific responsiveness represent a novel nanoplatform for simultaneous cell image and drug delivery.

## 1. Introduction

Bottlebrush polymers show an important kind of grafted copolymers with high molecular weights and various architectures (e.g., heterografted, core-shell, brush-on-brush, etc.) created by the “graft from”, “graft to”, and “graft through” approaches [1,2,3,4,5,6,7,8,9,10,11,12,13,14,15]. By integrating all kinds of backbone and side chains with multifarious functions, the properties and constructions of bottlebrush polymers could be manipulated easily for various desired applications. As one of the most investigated nanocarriers, amphiphilic copolymers with well-defined advanced topological structures of bottlebrush and the backbone of conjugated polymers with outstanding optical properties present both the ability of drug loading for therapy and cell imaging for diagnosis [16,17,18,19,20,21,22,23,24]. Polyfluorene (PF) and its derivatives with a superior fluorescence property, as well as chemical and thermal stability, emerges as one of the most investigated conjugated polymers for the fabrication of optical materials and biomaterials [25,26,27,28,29,30,31,32,33,34,35,36,37,38,39,40,41]. Wang’s ultrasmall theranostic unimolecular micelles assembled from the amphiphilic conjugated bottlebrush copolymer prodrugs constructed with the backbone of red fluorescent poly(fluorene-alt-(4,7-bis(hexylthien)-2,1,3-benzothiadiazole)) (PFTB) and side chains of hydrophobic disulfide have linked camptothecin and hydrophilic oligo(ethylene-glycol) for specific cancer treatment and in vivo bioimaging [41].

The hydrophilic corona of the nano-sized micelles assembled from amphiphilic copolymers could increase the solubility of the hydrophobic drugs loaded in the hydrophobic core in aqueous, prolong the blood circulation time, and enhance the utilization of these drugs via avoiding the foreign body response in vivo initiated by inflammatory cells through surface-absorbed proteins. The key point of the prolonged blood circulation time depends on the resistance of nonspecific protein adsorption to the surface hydration layer between the hydrophilic corona and water molecules [42,43,44,45]. As the most universal materials used to resist nonspecific protein adsorption, poly(ethylene glycol) (PEG) or oligo(ethylene glycol) (OEG) with antifouling characteristics up to its steric exclusion effect and hydration were widely investigated over the past decade [46,47]. In addition, because of equivalent positively and negatively charged groups, zwitterionic polymers with electrical neutrality present the better hydrophilicity and antifouling properties by forming a more stable and tighter hydration barrier through stronger electrostatic interaction of zwitterions, rather than hydrogen bonding of PEG/OEG, and universally serve for constructing nonbioadhesive interfaces of biomedical materials and devices with good plasma protein resistance [48,49,50,51,52]. Poly(sulfobetaine methacrylate) (PSBMA) with anionic sulfonate ion and cationic quaternary ammonium has been widely investigated because of its easy prepared synthesis [53,54,55,56,57,58,59,60]. Notably, Jiang’s stealth zwitterionic nanocarriers constructed by hierarchical self-assembly of two amphiphilic copolymers of poly(ortho ester)-block-poly(sulfobetaine methacrylate) and poly(ortho ester)-block-polysaccharide linked with a fluorescent dye of BODIPY, reveal the multifunctionality of prolonged circulation time, decreasing cargo leakage, and responsiveness of tumor environment to cancer therapy and bioimaging [61]. 

To our knowledge, zwitterionic conjugated bottlebrush copolymers integrating fluorescence imaging and tumor microenvironment-specific responsiveness for efficient intracellular drug release have been rarely reported, likely because of the lack of an efficient synthetic approach. We report in this study the successful preparation of well-defined theranostic zwitterionic conjugated bottlebrush copolymers with brush-on-brush architecture mimicking the supramolecular aggrecan aggregate structure. The copolymers possess compact construction and a lesser MW ratio between hydrophobic side chains and hydrophilic side chains. They are composed of a fluorescent backbone of PFONPN possessing the fluorescence resonance energy transfer with DOX, primary brushes of PHEMA, and secondary side chains of enzyme-degradable PTyr as well as zwitterionic P(OEGMA-*co*-SBMA) with super hydrophilicity and high antifouling ability via elegant integration of Suzuki coupling, NCA ROP, and ATRP techniques for significantly enhanced drug encapsulation and enzyme-triggered cleavage for accelerated intracellular drug release (Figure 1). Specifically, PFONPN-*g*-(PHEMA-*g*-(PTyr-*b*-P(OEGMA-*co*-SBMA)_2_)) and PFONPN-*g*-(PHEMA-*g*-(PTyr-*b*-POEGMA_2_)) with different constructions of hydrophilic brushes were synthesized (Figure 2 and Figure 3). The fluorescence property of the self-assembled micelles and the performance of DOX-loaded micelles for simultaneous cell imaging and in vitro anticancer drug delivery were investigated.

## 2. Results and Discussion

### 2.1. Polymer Synthesis and Characterization

The synthesis of zwitterionic bottlebrush conjugated copolymers featuring brush-on-brush architecture with compact construction composed of a fluorescent backbone of PFONPN, primary brushes of PHEMA, and secondary side chains of enzyme-degradable PTyr as well as zwitterionic P(OEGMA-*co*-SBMA) are showed in Figure 2 and Figure 3. The detailed reaction conditions, including amounts of initiator, monomer, catalyst, and reaction time, are summarized in Appendix A. The MW, degree of polymerization (DP), and polydispersity index (*Ð*) of all the synthesized polymers are summarized in Table 1. 

Firstly, the backbone of azide-functionalized polyfluorene (PFONPN-*g*-N_3_) and alkyne-PHEMA were synthesized by Suzuki coupling reaction and ATRP initiated by propargyl 2-bromoisobutyrate, respectively, followed by the click coupling between azide and alkyne. The successful synthesis of a parent conjugated polymer of azide-functionalized polyfluorene (PFONPN-*g*-N_3_) through Suzuki coupling reaction between monomer 1 and monomer 2 is confirmed by ^1^H-NMR as well as SEC-MALLS analyses (Appendix A and Figure 1a). The DP of PFONPN-*g*-N_3_ with a unimodal SEC elution peak and narrow distribution was determined to be approximately 9 on the basis of the MW determined by SEC-MALLS. Appendix A and Figure 1a present the typical ^1^H-NMR spectra and a unimodal SEC elution peak with a narrow distribution of alkyne-PHEMA, respectively. The DP of HEMA was calculated to be approximately 15 by the ratio of the integrated intensity of peak a (the methylene protons adjacent to alkynyl) and peak b or peak c (the methylene protons adjacent to hydroxyl) (Appendix A). The short chain of weaker hydrophilic alkyne-PHEMA removed easily by dialysis against distilled water was subsequently conjugated to the parent PFONPN-*g*-N_3_ with a 1:0.5 molar feed ratio of the azide group and the alkyne function, through a copper(I)-catalyzed azide–alkyne cycloaddition (CuAAC) click reaction to generate the multihydroxyl hydrophobic bottlebrush copolymer, PFONPN-*g*-PHEMA. The successful synthesis of PFONPN-*g*-PHEMA was confirmed by the appearance of all the characteristic signals of both PFONPN and PHEMA moieties (Appendix A). The high purity of the synthesized PFONPN-*g*-PHEMA was identified by the appearance of the resonance signal at 4.22 ppm attributed to the methylene protons adjacent to 1,2,3-1*H*-triazole in the ^1^H-NMR spectrum of PFONPN-*g*-PHEMA (Appendix A) as well as by the complete shift of the resonance signal at 4.65 ppm attributed to the methylene protons adjacent to alkynyl in the ^1^H-NMR spectrum of alkyne-PHEMA (Appendix A) to 5.03 ppm in the ^1^H-NMR spectrum of PFONPN-*g*-PHEMA (Appendix A) after click grafting and its narrowly distributed SEC elution peak (Figure 1a). 

Moreover, the ratio of the integrated intensity of peak a or peak b assigned to the methylene protons adjacent to 1,2,3-1*H*-triazole to that of peak a’ ascribed to the methylene protons adjacent to azide of PFONPN-*g*-PHEMA (Appendix A) was calculated to be 1/1.51, indicating 40% grafting of PHEMA.

Secondly, amino-functionalized conjugated bottlebrush macroinitiator PFONPN-*g*-(PHEMA-*g*-NH_2_) was synthesized by a CDI activation reaction between PFONPN-*g*-PHEMA and 1,6-hexamethylenediamine and then triggered the ring-opening polymerization (ROP) of Tyr-NCA. The successful introduction of amide groups into the termini of the pendant grafts of PFONPN-*g*-PHEMA, as the ring-opening polymerization (ROP) initiating the units for ROP of Tyr-NCA, is identified by the shift of the resonance signals at 3.57 ppm and 3.89 ppm attributed to the protons of the methylene adjacent to the hydroxyl group in the ^1^H-NMR spectrum of PFONPN-*g*-PHEMA (Appendix A) to 4.14 ppm (Appendix A) as well as the appearance of new resonance signals at 7.01 ppm and 1–1.5 ppm assigned to the amide and methylene of 1,6-hexamethylenediamine, respectively, in the ^1^H-NMR spectrum of PFONPN-*g*-(PHEMA-*g*-NH_2_) (Appendix A) after the reaction. Moreover, the ratio of the integrated intensity of peak c assigned to the protons of the amide of PFONPN-*g*-(PHEMA-*g*-NH_2_) to that of peak (a,b) ascribed to the protons of the ester group of PFONPN-*g*-(PHEMA-*g*-NH_2_) was calculated to be 1/3.87, indicating almost full decoration of the ROP-initiating sites with a high reaction efficiency. The DPs of PTyr brushes of PFONPN-*g*-(PHEMA-*g*-(PTyr-NH_2_)) were further calculated to be 16 by comparing the ratio of 1/1.96 of the integrated intensity of peak (b–e) at 1–1.5 ppm assigned to the methylene of 1, 6-hexamethylenediamine to that of peak h at 4.42 ppm assigned to the characteristic signals of PTyr grafts (Figure 2a).

Subsequently, conjugated bottlebrush ATRP macroinitiator PFONPN-*g*-(PHEMA-*g*-(PTyr-Br_2_)) was synthesized by the decoration of PFONPN-*g*-(PHEMA-*g*-NH_2_) with ATRP-initiating sites via imine links. The amino terminus in the pendant grafts of PFONPN-*g*-(PHEMA-*g*-(PTyr-NH_2_)) was subsequently converted to ATRP-initiating units for the final chain extension with POEGMA or P(OEGMA-*co*-SBMA) brushes through the reaction with the aldehyde group of CHO-Br_2_. The successful incorporation of ATRP-initiating sites is confirmed by the appearance of a new resonance signal (peak b) attributed to the methyl of the initiating units at 1.89 ppm in the ^1^H-NMR spectra of PFONPN-*g*-(PHEMA-*g*-(PTyr-Br_2_)) (Figure 2b) after the reaction. The ratio of the integrated intensity of peak b to that of peak a, attributed to the protons of tertiary carbon of PTyr, was calculated to be 1.30/1 for PFONPN-*g*-(PHEMA-*g*-(PTyr-Br_2_)), indicating almost complete decoration of the ATRP initiating sites. Although the SEC-MALLS reveals overlapped elution peaks for PFONPN-*g*-(PHEMA-*g*-NH_2_), PFONPN-*g*-(PHEMA-*g*-(PTyr-NH_2_)), and PFONPN-*g*-(PHEMA-*g*-(PTyr-Br_2_)) (Figure 1a) likely because of the strong absorption of amino-rich polymers with the SEC column, the UV signals recorded by UV detector indeed show clear shift as expected (Figure 1c).

Finally, target water-soluble conjugated bottlebrush copolymers, PFONPN-*g*-(PHEMA-*g*-(PTyr-*b*-P(OEGMA-*co*-SBMA)_2_)) and PFONPN-*g*-(PHEMA-*g*-(PTyr-*b*-POEGMA_2_)), with different constructions of hydrophilic polymer brushes, were synthesized by ATRP of OEGMA and SBMA initiated by the macroinitiator PFONPN-*g*-(PHEMA-*g*-(PTyr-Br_2_)). The copolymerization of OEGMA and SBMA is used to inhibit the hydrophobicity and aggregation caused by the interchain associations between side chains of PSBMA [62,63]. The DP of POEGMA and P(OEGMA-*co*-SBMA) brushes were calculated by comparing the integrated intensity of peak a attributed to the protons of the amide of PTyr brushes, peak (b,g) assigned to the characteristic signal of tertiary carbon of PTyr and the methylene protons adjacent to ester group of SBMA with that of peak d assigned to the methylene protons adjacent to ester group of OEGMA (Figure 2c). Because of the insolubility of SBMA in DMF, the SEC elution traces of the only three amphiphilic conjugated bottlebrush copolymers with merely POEGMA brushes or P(OEGMA-*co*-SBMA) with a small amount of SBMA show a detectable shift toward higher MW (Figure 1b) with similar Ð around 1.35. Notably, P_5_ with maximal MW showed the minimum shift toward higher MW, likely because of the strong absorption of SBMA with the SEC column.

### 2.2. Self-Assembly Behaviors of the Synthesized Zwitterionic Conjugated Bottlebrush Copolymers

Polymers with bottlebrush architectures have been repeatedly highlighted to form unimolecular micelles with greater stability than the ones formed by their traditionally linear analogs above the critical micelle concentration (CMC) [41,64,65,66,67]. The formation of unimolecular micelles of the synthesized zwitterionic conjugated bottlebrush copolymers was evaluated in dimethylsulfoxide (DMSO), a good solvent for all the moieties of the bottlebrush copolymers, and in water, the good solvent for the hydrophilic corona by dynamic light scattering (DLS), respectively. Accordingly, the mean size determined in DMSO by DLS reveals the practical dimension with free polymer chains, which should be larger than the mean size determined in water, accompanied by the collapse of the hydrophobic backbone and pendant inner brushes at the same polymer concentration but for the formation of self-assemble aggregates with larger dimension in the aqueous phase. P_3_ and P_4_ with the similar MW of ultralong and bulky hydrophilic blocks show a smaller mean size in water than that acquired in DMSO and keep apparent salt stability in HEPES (pH 7.4, 10 mM) with a little increase in dimension at the same polymer concentration of 0.5 mg/mL (Table 2). This indicates the formation of unimolecular micelles rather than aggregates from P_3_ and P_4_ in an aqueous phase, and the colloidal stability undisturbed by physiological salt conditions likely caused by the strong steric exclusion effect and hydration of pendant hydrophilic brushes. 

However, P_1_ with a shorter length of POEGMA self-assembled into aggregations in the aqueous phase, as well as in HEPES (pH 7.4, 10 mM) with significantly larger dimensions than that acquired in DMSO (Table 2), was probably a result of the finite hydration of shorter pendant hydrophilic brushes of POEGMA. In contrast, P_2_ modified with hydrophilic blocks of P(OEGMA-*co*-SBMA), with a similar MW to P_1_, could maintain the formation of unimolecular micelles not only in the aqueous phase but also in HEPES (pH 7.4, 10 mM) with the smaller mean size than that acquired in DMSO (Table 2), owing to the better hydrophilicity in forming a more stable and tighter hydration barrier through stronger electrostatic interaction of zwitterions rather than hydrogen bonding of PEG/OEG. In addition, the mean dimensions of P_1_ and P_2_ in a variety of diluted solutions were studied in water. The polymer concentration-independent mean size of P_2_ (Figure 3d, the corresponding mean sizes measured in DMSO are presented in Figure 3b) also clearly identifies the formation of unimolecular micelles, contrary to that of P_1_, with variational mean size influenced by the polymer concentration (Figure 3c, the corresponding mean sizes measured in DMSO are presented in Figure 3a). 

In addition, the larger dimensions of P_2_ and P_4_ in HEPES (pH 7.4) and especially in SSC (pH 7.4) result from the weaker electrostatic inter/intrachain dipole–dipole interaction and the stronger hydration associated with the antipolyelectrolyte property of polyzwitterionic brushes (Table 2), and might be beneficial for swelling and drug release of the drug-loaded micelles in acidic tumor environment [68,69,70,71]. The results imply a significant effect of polymer constructures on the stability of the formed self-assemblies in aqueous and physiological salt conditions. The morphological insight into the formation of aggregations of P_1_ or unimolecular micelles of P_2_ was further investigated by transmission electron microscopy (TEM) observation, identifying the well-dispersion of micelle nanoparticles with regularly spherical shape. The average diameter of P_1_ and P_2_ micelles at a polymer concentration of 0.5 mg/mL was estimated to be approximately 59 and 26 nm, respectively, in water from the TEM images (Figure 4c,d, the corresponding mean sizes determined by DLS are presented in Figure 4a,b), in accordance with the results investigated by DLS. The enzyme-triggered degradation of the PTyr brushes was further confirmed by incubating polymer micelles with proteinase K (6 U mL^−1^) for 12 h at 37 °C, leading to an obvious shift from the unimodal size distribution to multiple size populations, with a very board size distribution for both P_1_ and P_2_-based polymer micelles (Figure 5c,d), which is expected to promote intracellular drug diffusion and release for enhance therapeutic efficiency.

### 2.3. In Vitro Drug Loading and Drug Release Study

Doxorubicin (DOX) was chosen to evaluate the performance of synthesized water-soluble bottlebrush copolymers P_2_–P_4_ in terms of anticancer drug delivery. DOX-loaded micelles were prepared by a classical dialysis method. The drug-loading content (DLC) and encapsulation efficiency (EE) of P_2_–P_4_ micelles with a different feed ratio of DOX and polymer are summarized in Table 3. Micelles of DOX@P_2_-2 show a greater DLC and EE than DOX@P_3_ and DOX@P_4_ analogs with the same feed ratio of DOX, on account of the overstretching of PTyr blocks from the ultralong and bulky hydrophilic blocks of P_3_ and P_4_. All the drug-loaded micelles show a slightly greater average size than the blank micelles in HEPES because of the encapsulation of drug molecules within the hydrophobic core and the hydration barrier of the micelles (Table 3). 

Transmission electron microscopy (TEM) observation also shows the presence of well-dispersed micelles with a regularly spherical shape after drug loading, identifying the excellent colloidal stability of the unimolecular micelles formed by the zwitterionic bottlebrush copolymers impregnable from drug incorporation. Specifically, the average diameter of DOX@P_2_-1 and DOX@P_2_-2 micelles at a polymer concentration of 0.5 mg mL^−1^ was estimated to be approximately 35 and 43 nm, respectively, in water from the TEM images (Figure 6c,d; the corresponding mean sizes determined by DLS are presented in Figure 6a,b).

The in vitro DOX release profiles of DOX@P_2_-1 micelles were further investigated at 37 °C under the physiological conditions (HEPES, pH 7.4, 10 mM), imitating the typical extracellular pH in the presence or absence of proteinase K (6 U mL^−1^), as well as in an acidic medium (SSC, pH 5.0 150 mM) mimicking the tumor intracellular acidic pH, respectively. Incubation of DOX@P_2_-1 micelles in the physiological condition of pH 7.4 led to 28% drug release in 96 h, which is much lower than the cumulative drug release acquired in the same release period in HEPES (pH 7.4, 10 mM) (69%) with proteinase K (6 U mL^−1^) and in an acidic medium of pH 5.0 (55%). This was due substantially to the enzyme-triggered degradation of PTyr brushes for micelle degradation and accelerated drug release and protonation of the glycosidic amine for increased solubility of DOX in an acidic medium, respectively (Figure 7).

### 2.4. Photophysical Properties

The optical properties of P_1_–P_4_, as well as their DOX-loaded micelles, were further studied, including the fluorescence excitation and emission spectra and fluorescence quantum yields (Table 4). 

Take P_2_, for example, the synthesized water-soluble bottlebrush copolymer shows the characteristic emission peak at 520 nm in DMSO and 490 nm in water at an excitation wavelength of 390 nm, and the conjugated backbone of PFONPN-*g*-N_3_ presents the characteristic emission peak at 540 nm in DMSO at an excitation wavelength of 420 nm (Figure 8a,b). The apparent shift of the characteristic excitation and emission peak of bottlebrush polymer compared with that of the backbone of PFONPN-*g*-N_3_ might be related to the presence of abundant pendant phenol groups on the side chains of PTyr blocks. Interestingly, the excitation peak of DOX fully overlaps with the emission peak of P_2_, probably leading to the fluorescence resonance energy transfer (FRET) from the PFONPN backbone to DOX (Figure 8b). To study the FRET from the PFONPN backbone of P_2_ to DOX, fluorescence emission spectra of P_2_ added with a different feed ratio of DOX with the same concentration of 0.5 mg/mL were detected with excitation at 390 nm in water (Figure 8c,d). As expected, the emission spectrum of DOX was detected with an excitation wavelength of 390 nm that was used for the PFONPN moiety based on the FRET from the PFONPN backbone to DOX. The fluorescence intensity of DOX at 600 nm increases remarkably, accompanied by the decrease in fluorescence intensity of PFONPN and the increase in the concentration of DOX from 0–6% (Figure 8c). However, the fluorescence intensity in DOX starts to drop with the increase in the concentration of DOX from 7 to 10% because of the fluorescence quenching of DOX in high concentration (Figure 8d).

The presence of brushes in the bottlebrush copolymers has been highlighted to prevent the aggregation of the conjugated backbone without compromised fluorescence quantum yields, as in previous research. To investigate this structure–property relationship, the fluorescence quantum yields of P_1_–P_4_ were detected in water and DMSO with an excitation wavelength of 390 nm. However, the fluorescence quantum yield of these water-soluble bottlebrush copolymers in water and DMSO is significantly lower (32.07%) than that of the conjugated backbone of PFONPN-*g*-N_3_ in DMSO, probably because of the presence of abundant pendant phenol groups on the side chains of PTyr blocks for increased occurrence of nonradiative pathways in the relaxation of excited states of PFONPN.

### 2.5. In Vitro Cellular Uptake and Cytotoxicity

In vitro cellular uptake of blank micelles of P_2_ and drug-loaded micelles of DOX@P_2_-1 were measured by fluorescence microscopy with the control of free DOX. DAPI was used to stain the nuclei with blue fluorescence. After incubated with P_2_ and DOX@P_2_-1, Bel-7402 cells present green and red fluorescence throughout cellular cytoplasm derived from PFONPN and DOX, respectively, as well as the merged image throughout the perinuclear region, revealing the successful intracellular transportation and excellent abilities for cell imaging of the brush-on-brush copolymers (Figure 9). Flow cytometry analysis was applied to quantify in vitro cellular uptake efficiency of P_2_ and DOX@P_2_-1 (Figure 10a). Because of different internalization mechanisms between free DOX with fast membrane permeation and polymer constructs with slow endocytosis, free DOX shows obvious higher cellular uptake efficiency compared with DOX-loaded micelles. 

In vitro cytotoxicity of P_2_ and DOX@P_2_-1 was investigated by MTT cell viability assay. The blank micelles of P_2_ are nearly nontoxic to L02 cells and Bel-7402 cells, with cell viability of 80% after being incubated with P_2_ at the concentration of 64 μg/mL (Figure 10b). The DOX-loaded micelles of DOX@P_2_-1 reveal significant cytotoxicity to L02 cells (Figure 10c) and Bel-7402 cells (Figure 10d), with the half-maximal inhibitory concentration (IC_50_) of 7.0 (3.4, 8.6) μg/mL and 10.6 (7.4, 16.6) μg/mL, respectively. The less cytotoxic activity of DOX@P_2_-1 compared with free DOX is likely attributed to the slower internalization mechanism and release kinetics of DOX from polymeric micelles. The overall results present the significant ability of the “theranostic” micelles based on the brush-on-brush copolymers in the field of simultaneous cellular imaging and drug delivery.

## 3. Materials and Methods

### 3.1. Materials and Characterizations

Copper(I) bromide (CuBr) and oligo (ethylene glycol) monomethyl ether methacrylate (OEGMA, Mn = 300 g/mol) and 2-hydroxyethyl methacrylate (HEMA) were purchased from Sigma-Aldrich (Shanghai, China). OEGMA and HEMA were passed through a basic alumina column to remove the inhibitor. Triethylamine (TEA), 2,2′-Bipyridine (Bpy), 4-dimethylaminopyridine (DMAP), 2-bromoisobutyryl bromide, *N,N*′-carbonyldiimidazole (CDI), 1,6-hexamethylenediamine, 2,7-dibromofluorene, 1,6-dibromohexane, tetrabutylammoniumbromide (TBAB), 2-(methacryloyloxy)ethyl]dimethyl-(3-sulfopropyl) ammonium hydroxide (SBMA), and bis(pinacolato)diborane were purchased from J&K (Beijing, China), and used as received without further purification. Triphosgene, l-tyrosine (l-Tyr), 4-bromo-1,8-naphthalic anhydride,4-bromoaniline, 4-formylbenzoic acid, 1,1,1-tris(hydroxymethyl)ethane, *N,N*′-dicyclohexylcarbodiimide (DCC), CsCO_3_, doxorubicin hydrochloride (DOX·HCl), proteinase K and (Pd(dppf)Cl_2_) (dppf = 1,1′-bis (diphenylphosphanyl) ferrocene)) were purchased from Aladdin Sodium azide (NaN_3_, Sanyou, Shanghai, China). Tetrahydrofuran (THF), dimethyl sulfoxide (DMSO), *N,N*-dimethylformamide (DMF), and other reagents were used as received without further purification. 4-Bromo-9-*N*-4-bromophenyl-1,8-naphthalimide [72], 2,7- bis(4,4,5,5-tetramethyl-1,3,2-dioxaborolan-2-yl)-9,9-bis(6′-azidohexyl) fluorine [73], propargyl 2-bromoisobutyrate [74], Tyr-NCA [75], and 2,2-(bis(2-bromo-2-methylpropanoyl)oxy)propyl 4-formylbenzoate (CHO-Br_2_) [75] were synthesized according to the reported procedures.

^1^H-NMR spectra were measured on a JNM-ECS 400 MHz NMR spectrometer (JEOL, Tokyo, Japan). The molecular weight (MW) and molecular weight distribution (*Ð*) were determined by the size exclusion chromatography and multiangle laser light scattering (SEC-MALLS) with the eluent of HPLC-grade DMF containing 0.1 wt% LiBr at 60 °C and a flow rate of 1 mL/min. Tosoh TSK-GEL R-3000 and R-4000 columns (Tosoh Bioscience, Shanghai, China) were connected in series to an Agilent 1260 series (Agilent Technologies, Santa Clara, CA, USA), an interferometric refractometer (Optilab-rEX, Wyatt Technology, Goleta, CA, USA), and a MALLS device (DAWN EOS, Wyatt Technology, Goleta, CA, USA). The MALLS detector was operated at a laser wavelength of 690.0 nm. Particle size and size distribution were measured on Zetasizer (Nano ZS, Malvern, Worcestershire, UK) with the detection angle fixed at 173°. The TEM images were recorded on a JNM-2010 instrument (JEOL, Tokyo, Japan) operating at an acceleration voltage of 200 keV. The samples were stained using phosphotungstic acid (1% *w*/*w*) and dried in the air prior to visualization. Fluorescence spectra and fluorescence quantum yields were recorded on a Steady-State, Time-Resolved Fluorescence Spectrofluorometer (Edinburgh Instruments, England). The excitation wavelength was fixed at 390 nm. The emission spectra were recorded from 400 to 650 nm. The bandwidths of excitation and emission were both 5 nm.

### 3.2. Synthesis of 4-Bromo-9-N-4-bromophenyl-1,8-naphthalimide (1)

4-bromo-1,8-naphthalic anhydride (2.89 g, 10 mmol), 4-bromoaniline (5.21 g, 30 mmol), and ethyl alcohol (50 mL) were added to a 100 mL three-necked flask, stirred vigorously, and refluxed overnight. After cooling to room temperature, the precipitates were collected by filtration, followed by wishing with ethyl alcohol, recrystallized from acetone, and desiccated in a vacuum to afford a yellow solid (3.10 g, yield: 72%):^1^H-NMR (400 MHz, CDCl_3_), δH [ppm]: 8.61 (m, 2H), 8.36 (t, 1H), 8.27(m, 1H), 8.05 (m, 1 H), 7.74 (d, 2 H), and 7.39 (d, 2 H).

### 3.3. Synthesis of 2,7-Bis(4,4,5,5-tetramethyl-1,3,2-dioxaborolan-2-yl)-9,9-bis(6′-azidohexyl) Fluorine (2)

Monomer 2 was synthesized according to the procedures previously reported for the synthesis of amphiphilic-conjugated bottlebrush copolymers, PF-((*g*-PCL-OOCCH_3_)-*alt*-(*g*-POEGMA)) [73].

### 3.4. Synthesis of Azido-Functionalized Polyfluorene (PFONPN-g-N_3_) by Suzuki Coupling Reaction

To a mixture solution of THF (10 mL) and water (1.5 mL), monomer 1 (0.216 g, 0.5 mmol), monomer 2 (0.334 g, 0.5 mmol), (Pd(dppf)Cl_2_) (10 mg), CsCO_3_ (0.65 g, 2 mmol), and Alquat 336 (0.016 mg, 0.04 mmol) was added. After three freeze-pump-thaw cycles to replace air with nitrogen, the mixture was kept for 1 h at 85 °C. After the reaction, the mixture was added dropwise into cold methanol (150 mL) to precipitate the product, which was later harvested by centrifugation. The precipitates were redissolved in DMF, followed by filtration. The filtered solution was dialyzed against distilled water. The product was harvested by centrifugation and freeze-drying to afford a yellow solid (154 mg, yield: 45%).

### 3.5. Synthesis of Propargyl 2-Bromoisobutyrate

Triethylamine (3.36 mL, 24 mmol) and propargyl alcohol (1.12 g, 20 mmol) were dissolved in 50 mL of anhydrous DCM and stirred in a 100 mL flame-dried round bottom flask at 0 °C. Then, 2-bromoisobutyryl bromide (3.02 mL, 24 mmol) in 15 mL of anhydrous DCM was added dropwise to the above solution at 0 °C while stirring. After addition, the reaction mixture was stirred at room temperature for 12 h. The solution was filtered, the filtered solution was washed with water four times, and the solution was dried with Na2SO4. After evaporating the solvent by rotary evaporation, the crude product was purified by column chromatography over silica gel eluting with hexane/ethyl acetate (20:1) to give the product an appearance of a colorless oil-like product (3.12 g, yield: 76%): ^1^H-NMR (400 MHz, CDCl_3_), δH [ppm]: 4.77 (d, 2H), 2.51 (t, 1 H), and 1.96 (s, 6H).

### 3.6. Synthesis of Alkyne-PHEMA by ATRP

HEMA (390 mg, 3 mmol), propargyl 2-bromoisobutyrate (41 mg, 0.2 mmol), and Bpy (63 mg, 0.4 mmol) were dissolved in a 9:1% *w*/*w* IPA/DMF mixture to obtain a 50% *w*/*w* HEMA solution. After three freeze-pump-thaw cycles, CuBr (29 mg, 0.2 mmol) was then added quickly under the protection of nitrogen flow. After another three freeze-pump-thaw cycles, the reaction mixture was kept at 65 °C for 20 min. After the reaction was completed, the solution was quenched by exposure to the air and diluted with DMF, followed by precipitation in cold anhydrous diethyl ether to obtain the crude product. The crude product was purified by dialysis against distilled water to remove copper catalyst and any unreacted monomer. The purified product was obtained by freeze-drying to afford a white solid (289 mg, yield: 67%).

### 3.7. Synthesis of Conjugated Bottlebrush Copolymers PFONPN-g-PHEMA by Click Reaction through “Grafting to” Approach

PFONPN-g-N_3_ (48 mg, 0.14 mmol –N_3_), alkyne-PHEMA (151 mg, 0.07 mmol), PMDETA (13.48 mg, 0.077 mmol), and CuBr_2_ (1.58 mg, 0.007 mmol) were dissolved in DMF (5 mL). After three freeze−pump−thaw cycles, CuBr (10.04 mg, 0.07 mmol) was loaded under the protection of nitrogen flow. After another three freeze−pump−thaw cycles, the mixture was sealed and kept at 45 ℃ for 12 h. The reaction was quenched by exposure to air, followed by dialysis against distilled water. Because of the hydrophobicity of the product, the formation of precipitates was observed in the dialysis tube. The precipitates were harvested by centrifugation and freeze-drying to afford a pale-yellow solid (159 mg, yield: 80%).

### 3.8. Synthesis of Tyr-NCA and CHO-Br_2_

Tyr-NCA and CHO-Br_2_ were synthesized according to the procedures previously reported for the synthesis of enzyme-responsive theranostic amphiphilic bottlebrush copolymers, PF-*g*-(PTyr-*b*-(POEGMA)_2_ [75].

### 3.9. Synthesis of Amino-Functionalized Conjugated Bottlebrush Copolymers PFONPN-g-(PHEMA-g-NH_2_)

PFONPN-*g*-(PHEMA) (100 mg, 0.8 mmol -OH) and CDI (1.32 g, 8 mmol) were dissolved in dry THF (30 mL), followed by stirring overnight at room temperature. THF (100 mL) was added to the reaction mixture, which was then washed five times with saturated brine solution, dried over anhydrous MgSO_4_, and filtered to obtain the imidazole-polyfluorene (PFONPN-*g*-(PHEMA-CDI)) solution.

1,6-Hexamethylenediamine (1.87 g, 16 mmol) and DMAP (10 mg, 0.08 mmol) were added to dry THF (100 mL). After stirring for 5 min, all starting materials were dissolved, and the PFONPN-*g*-(PHEMA-CDI) solution was added dropwise, followed by stirring at room temperature for 24 h. After completion, the reaction mixture was concentrated by vacuum evaporation and further dissolved in THF, then added dropwise into excess ice-cold Et_2_O to precipitate the crude product. The precipitates were redissolved in DMF and dialyzed against distilled water. The product was harvested by centrifugation and freeze-drying to afford a yellow solid (177 mg, yield: 81%)

### 3.10. Synthesis of Conjugated Bottlebrush Copolymers PFONPN-g-(PHEMA-g-(PTyr-NH_2_)) by Ring-Opening Polymerization through “Grafting from” Approach

As an initiator, PFONPN-*g*-(PHEMA-*g*-NH_2_) (50 mg, 0.18 mmol –NH_2_) was quickly added to a solution of Tyr-NCA (1.04 g, 5 mmol) in DMF (10 mL) under an N_2_ atmosphere. The reaction proceeded at 45 °C for 24 h, then added dropwise into excess ice-cold Et_2_O to precipitate the crude product. The precipitates were redissolved in DMF, followed by dialysis against DMF and deionized water for 24 h, respectively. The product was harvested by centrifugation and freeze-drying to afford a white solid (546 mg, yield: 63%).

### 3.11. Synthesis of Conjugated Bottlebrush ATRP Macroinitiator PFONPN-g-(PHEMA-g-(PTyr-Br_2_))

PFONPN-*g*-(PHEMA-*g*-(PTyr-NH_2_)) (290 mg, 0.1 mmol –NH_2_) and a few anhydrous MgSO_4_ were added to the solution of CHO-Br_2_ (550 mg, 1 mmol) in dry DMF (10 mL) under an N_2_ atmosphere. The reaction proceeded at 80 °C for 24 h, then added dropwise into excess ice-cold Et_2_O to precipitate the crude product. The precipitates were redissolved in DMF and dialyzed against DMF and deionized water for 24 h, respectively. The product was harvested by centrifugation and freeze-drying to afford a pale-yellow solid (272 mg, yield: 79%).

### 3.12. Synthesis of Zwitterionic Conjugated Bottlebrush Copolymers PFONPN-g-(PHEMA-g-(PTyr-b-P(OEGMA-co-SBMA)_2_)) by ATRP through “Grafting from” Approach

For P_5_, for example, PFONPN-*g*-(PHEMA-*g*-(PTyr-Br_2_)) (35 mg, 0.02 mmol –Br), Bpy (6 mg, 0.04 mmol), SBMA (171 mg, 0.6 mmol), and OEGMA (420 mg, 1.4 mmol) were dissolved in DMSO (10 mL) and H_2_O (1 mL). After three freeze-pump-thaw cycles, CuBr (3 mg, 0.02 mmol) was loaded under the protection of nitrogen flow. After another three freeze-pump-thaw cycles, the mixture was sealed and kept at 60 °C for 1 h. The reaction mixture was quenched by being exposed to air and dialysis against distilled water to remove any unreacted monomer and copper catalyst. The purified product was harvested by freeze-drying.

### 3.13. In Vitro Drug Loading and Drug Release

For DOX@P_2_-1, for example, DOX·HCl (5 mg) and TEA (50 μL) were added to 2 mL of DMSO and stirred overnight in the dark at room temperature. P_2_ (50 mg) dissolved in 2 mL of DMSO was added to the above solution and stirred for 1 h at room temperature, followed by the addition of 4 mL ultrapurified water, drop by drop, under vigorous stirring for another 1 h. The mixture was dialyzed against distilled water for 24 h. The drug-loaded micelle was collected by freeze-drying.

The in vitro drug release study of drug-loaded micelles was investigated, respectively, in HEPES (10 mM, pH 7.4), saline sodium citrate (SSC, pH 5.0, 150 mM), and HEPES (10 mM, pH 7.4) containing proteinase K (6 U mL^−1^) at 37 °C. The drug-loaded micelles were redispersed in buffer solution with a concentration of 0.5 mg/mL; 1 mL of the solution was added to the dialysis bag (MWCO 12,000 kDa) and immersed in a tube loaded with 30 mL of release medium, then placed in a horizontal laboratory shaker thermostated with stirring speed of 120 rpm at 37 °C. The release medium (3 mL) was taken out, and a fresh medium with equal volume was added at the predetermined time intervals. The drug concentration was calculated by a UV−Vis spectrometer.

### 3.14. In Vitro Cellular Uptake

Cell imaging was investigated by a BioTek LIONHEART/LX automated microscope. Bel-7402 cells were seeded with the density of 5 × 10^5^ cells per well and incubated in a 37 °C, 5% CO_2_ environment for 24 h. The solutions in a complete growth medium with a concentration of 0.25 μg DOX equv./mL (25% of the IC_50_ of free DOX) were added to the wells and incubated for 4 h at 37 °C. Then, cells were fixed with 4% paraformaldehyde (PFA) solution for 10 min, stained with 2-(4-amidinophenyl)-6-indolecarbamidine (DAPI), and imaged by microscope.

The quantificational evaluation of cellular uptake was further studied by flow cytometry. Bel-7402 cells were seeded with the density of 1 × 10^5^ cells per well and incubated in a 37 °C, 5% CO_2_ environment for 24 h. The solutions in a complete growth medium with the concentration of 0.25 μg DOX equv./mL were added to the wells and incubated for 4 h at 37 °C. Cells were rinsed twice with PBS and harvested by incubation with 200 μL of Trypsin-EDTA, followed by resuspension with 1 mL of complete growth medium. Then, the cell suspension was transferred to 1.5 mL microcentrifuge tubes and pelleted for 5 min. The supernatant was removed, and the cells were resuspended in 200 μL of PBS. Cells were measured quantitatively for uptake of DOX in a free or encapsulated state with an excitation wavelength and emission wavelength of 488 nm and 595 nm (20,000 cells counted).

### 3.15. Cell Viability Study

The cytotoxicity was investigated by MTT assay. The Bel-7402 or L02 cells were seeded with a density of 1 × 10^4^ cells per well and incubated for 12 h. Samples were dispersed in a cell culture medium with different concentrations and added to the wells with 100 μL. After incubating for 24 h, MTT solution (20 μL, 5 mg/mL) was added and incubated for another 4 h. The absorbance was measured at 490 nm by a Bio-Rad microplate reader (BD Biosciences, San Jose, CA, USA).

### 3.16. Statistical Analysis

Data are recorded as the mean ± standard deviation by one-way analysis of the Student’s *t*-test. The Statistical differences were defined as significant for *** *p* < 0.001.

## 4. Conclusions

In summary, we developed the successful preparation of enzyme-responsive theranostic zwitterionic bottlebrush copolymers featuring brush-on-brush architecture and consisting of a fluorescent backbone of PFONPN possessing fluorescence resonance energy transfer with DOX, primary brushes of PHEMA, and secondary graft brushes of enzyme-degradable side chains of PTyr as well as zwitterionic side chains of P(OEGMA-*co*-SBMA) by elegantly integrating various techniques of Suzuki coupling, NCA ROP, and ATRP. The resulting brush-on-brush copolymer with tumor microenvironment-specific responsiveness could form unimolecular micelles in aqueous with high drug-loading content because of the super-hydrophilic zwitterionic brushes, and it represents a novel nanoplatform for cancer theranostics. 

## Data Availability

This study did not report any data.

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
