# Peer review of "Construction of Enzyme-Responsive Micelles Based on Theranostic Zwitterionic Conjugated Bottlebrush Copolymers with Brush-on-Brush Architecture for Cell Imaging and Anticancer Drug Delivery"

_molecules, 2022, doi:10.3390/molecules27093016_

Round 1
Reviewer 1 Report
The manuscript describes an efficient method for the synthesis of enzyme-responsive theragnostic micelles for anti-cancer drug delivery and imaging.
Major issues:
The design of PFONPN for cell imaging is not reasonable. The green light background in vivo is higher than those lights with longer wavelengths, such as red, violet even far violet and the light can be easily replaced in the present design. In addition, to study the actual application, the in vivo imaging and drug delivery efficiency should be tried.
Minor issues:
Fig.9, X axle should be DAPI, PRONPN, Dox.
Fig.5, Panel c and d are not explained in the figure caption.
Fig.4, the sizes of P1 and P2 TEM images are not consistent with the graphs.
Fig.10, Y axle, spelling error for fluorescence.
Fig. 11, the panel a was wrongly drawn and x axle should be text. Statistical comparison is also required.
Line 325, ae to as.
Line 602, the study did not report any data?
Abbreviation issues. MTT, HPLC, etc lack full names.
Spelling issues. “-1” in “ML-1” should be superscript. Check other similar issues throughout the text tables, and figs.
Reviewer 2 Report
The authors report the synthesis of zwitterionic conjugated bottlebrush copolymers and the resulting micelles loaded with doxorubicin for cell imaging and drug delivery. The paper is well written and organized and the reported results have relevance in biomedical field. However, I believe that before the manuscript can be considered for publication in this Journal, the novelty of the work must be addressed; the same research group have already reported very similar results in “F. Liu et al., Fabrication of theranostic amphiphilic conjugated bottlebrush copolymers with alternating heterografts for cell imaging and anticancer drug delivery, Polym. Chem., 2018,9, 4866-4874”. This reference was not reported in the manuscript. Moreover, what are the advances and future directions of the reported research?
Round 2
Reviewer 2 Report
I believe that this revised version of the manuscript can be accepted for publication in this Journal.